# Micrococcin P1 and P2 from Epibiotic Bacteria Associated with Isolates of *Moorea producens* from Kenya

**DOI:** 10.3390/md20020128

**Published:** 2022-02-07

**Authors:** Thomas Dzeha, Michael John Hall, James Grant Burgess

**Affiliations:** 1D. John Faulkner Centre for Marine Biodiscovery and Biomedicine, P.O. Box 4, Kinango 80405, Kenya; 2Department of Pure and Applied Sciences, Technical University of Mombasa, P.O. Box 90420, Mombasa 80100, Kenya; 3School of Natural and Environmental Sciences, Newcastle University, Newcastle upon Tyne NE1 7RU, UK; michael.hall@newcastle.ac.uk (M.J.H.); grant.burgess@newcastle.ac.uk (J.G.B.)

**Keywords:** *Moorea producens*, CuSO_4_·5H_2_O assisted, differential gDNA isolation, filamentous bacteria, micrococcin P1 and P2, stalked diatoms

## Abstract

Epibiotic bacteria associated with the filamentous marine cyanobacterium *Moorea producens* were explored as a novel source of antibiotics and to establish whether they can produce cyclodepsipeptides on their own. Here, we report the isolation of micrococcin P1 (**1**) (C_48_H_49_N_13_O_9_S_6_; obs. *m*/*z* 1144.21930/572.60381) and micrococcin P2 (**2**) (C_48_H_47_N_13_O_9_S_6_; obs. *m*/*z* 1142.20446/571.60370) from a strain of *Bacillus marisflavi* isolated from *M. producens*’ filaments. Interestingly, most bacteria isolated from *M. producens*’ filaments were found to be human pathogens. Stalked diatoms on the filaments suggested a possible terrestrial origin of some epibionts. CuSO_4_·5H_2_O assisted differential genomic DNA isolation and phylogenetic analysis showed that a Kenyan strain of *M. producens* differed from *L. majuscula* strain CCAP 1446/4 and *L. majuscula* clones. Organic extracts of the epibiotic bacteria *Pseudoalteromonas carrageenovora* and *Ochrobactrum anthropi* did not produce cyclodepsipeptides. Further characterization of 24 Firmicutes strains from *M. producens* identified extracts of *B. marisflavi* as most active. Our results showed that the genetic basis for synthesizing micrococcin P1 (**1**), discovered in *Bacillus cereus* ATCC 14579, is species/strain-dependent and this reinforces the need for molecular identification of *M. producens* species worldwide and their epibionts. These findings indicate that *M. producens*-associated bacteria are an overlooked source of antimicrobial compounds.

## 1. Introduction

Global health care is continually facing new challenges from human pathologies including cancers, diseases of old age (e.g., Alzheimer’s and Parkinson’s), rare orphan diseases, emerging infectious diseases, and, increasingly, the threat of drug-resistant infectious diseases. Infectious diseases claim over 13 million lives worldwide each year, with over 700,000 deaths currently attributed to antibiotic-resistant infections [1]. Drug resistance threatens to overturn the gains in antibiotic discovery since penicillin. Thus, there is an overwhelming urgency to discover new antibiotics, especially to fight antibiotic-resistant strains of MRSA, *Pseudomonas aeruginosa*, and *Klebsiella pneumoniae*. New natural products are needed not only to address antibiotic resistance, but also the aforementioned healthcare challenges.

To date, bacteria have provided the majority of the antimicrobial agents so far discovered [2,3]. Historically, the main source of bacteria for isolation of antibiotics has been soils [4] and, more recently, marine sediments [5,6]. Spore-forming actinomycetes that are defined as “chemically diverse bacteria” due to their ubiquitous production of antibiotics are abundant in soils but are also found in the marine environment. It is uncertain if they originate in the ocean or are washed into the sea in large numbers [6,7]. The study of bacteria that live on the surfaces of other organisms or symbiotically within other organisms has also been fruitful [8,9]. There is evidence that some organisms such as seaweeds and cyanobacteria harbor interesting holobionts, which are capable of producing bioactive compounds [10]. Cyanobacteria are also well known as sources of a diverse range of bioactive compounds. On the other hand, the study of cyanobacteria-associated bacteria in the discovery of novel antibiotics is in its infancy.

Bacteria–cyanobacteria associations are pre-historic at 440 Ma years ago, according to fossil evidence [11]. Bacteria attach themselves to cyanobacteria for buoyancy and to obtain nutrients released during photosynthesis; there is often a symbiotic relationship between these prokaryotes. Cyanobacterial filaments can be surrounded by a tough polysaccharide sheath that can harbor a multitude of heterotrophic bacteria that are difficult to remove [12]. The dominating heterotrophic bacteria associated with these sheaths are Gram-negative species from the phyla Proteobacteria and Bacteroidetes [13].

The association of bacteria with a cyanobacterial host is quite diverse and encompasses all phyla including Firmicutes, Actinobacteria, and γ-proteobacteria [14]. These assemblages constitute highly specific symbiotic relationships. Four specific strains of highly colored bacteria producing a quinone antibiotic were associated with *Lyngbya majuscula* from Puerto Rico [15]. The isolation of two minor hydroxyquinone compounds, quinone alkaloid 1 and 2 from a Curacao strain of *L. majuscula*, led to the speculation that these compounds may be synthesized by associated bacteria [16]. In a later study, the filamentous non-heterocystous marine cyanobacteria *Lyngbya* sp. was shown to harbor a maximum of six isolates of cultivable bacteria [12]. A study of other species of the order Oscillatoriales, comprising *Oscillatoria pseudogeminata*, *O. subtilissima*, *O. amphibian*, *O. cortiana*; *Phormidium lucidum*, *Phormidium* sp., *P. valderianum*, *P. tenue*, *P. foveolarum*; *Lyngbya confervoides* BDU 140301, *Lyngbya* sp. BDU 91711, and *Lyngbya* sp. BDU 141561, led to the isolation of 46 cultivable bacterial isolates [12].

The filamentous marine cyanobacterium *Moorea producens* is known to grow symbiotically with a number of other microorganisms, some of which are difficult to isolate [17]. *M. producens* has also been linked with swimmers’ itch arising from debromoaplysiatoxin (DAT) [18]. In common with other Oscillatoriales, a Kenyan isolate of the filamentous marine cyanobacteria *M. producens* has associated bacteria including those with the ability to resist UV radiation [19]. Taxonomically, Moorea has recently been renamed Moorena [20]. This cyanobacterium is also a source of the cyclodepsipeptides homodolastatin 16 and antanapeptin A [21] and the potent anticancer molecule, dolastatin 16 (this study). The quest for a sustainable supply of these compounds generated an interest in obtaining the metagenomic DNA (gDNA) of the cyanobacteria/bacterial community. The immense diversity of the *Oscillatoriales* suggests that morphological identification is unreliable [13]. By contrast, molecular identification is accurate and specific as it is based on the genomic content of the organism. Whereas the latter method works efficiently for axenic species, its use for the speciation of non-axenic cyanobacteria is inadequate due to intragenomic gene heterogeneity arising from the presence of bacteria and other microorganisms [13,22,23]. Currently, molecular identification of non-axenic cyanobacteria utilizes the multiple displacement amplification (MDA) method, which has a limited total genomic coverage [24,25].

Various approaches for obtaining axenic cultures of cyanobacteria from non-axenic strains are documented [26]. However, they do not elaborate on how to isolate gDNA from non-axenic strains. Bacteria, yeast, and viruses are easily killed on contact with Cu^2+^ ions by reactive hydroxyl ions and hydroxy radicals that damage the cell while denaturing DNA in a Fenton-type reaction [27].
Cu^+^ + H_2_O → Cu^2+^ + OH^−^ + OH^●^

The use of copper and its alloys in “contact” killing of microorganisms dates back to both the Sumerian and Akkadian civilizations and only lost its prominence in medicine at the advent of commercially available antibiotics in 1932 [28]. However, the contact killing of microbes by copper is still being applied in Agriculture to control bacterial and fungal diseases [27]. Thus, we report here a novel method for the CuSO_4_·5H_2_O assisted differential gDNA isolation of non-axenic filamentous marine cyanobacteria. It is based on the treatment of cyanobacterial biomass with CuSO_4_·5H_2_O prior to exhaustive differential isolation of bacterial gDNA from cyanobacterial biomass to provide a residual substrate for isolation of the cyanobacteria gDNA. CuSO_4_·5H_2_O provides the Cu^2+^ ions needed for the contact killing of bacteria on the cyanobacteria biomass during treatment. A Kenyan isolate of *M. producens* was used for these studies. In addition, this method was used to confirm the identity of *L. majuscula* CCAP 1446/4 from the Culture Collection, Oban, Scotland, and to delineate the Kenyan *M. producens* from other species elsewhere. Bacteria associated with *M. producens*, including those associated with the filaments of *M. producens*, are described. The current study also highlights the first isolation of the antibiotics micrococcin P1 (**1**) and micrococcin P2 (**2**) from organic extracts of *B. marisflavi* and establishes an IC_50_ value of micrococcin P1 (**1**) against *Staphylococcus aureus*.

## 2. Results and Discussion

### 2.1. Detection of Homodolastatin 16, Dolastatin 16, and Antanapeptin A in Kenyan Isolates of a Filamentous Cyanobacterium, Moorea producens

Besides morphological appearance, the identity of the filamentous Kenyan marine cyanobacterium *M. producens* [21] was supported by the detection of a number of natural products known to be associated with this species. Following organic extraction of the sample and fractionation on C-18 silica, comparative HRFABMS against a dolastatin 15 standard, showed the presence of homodolastatin 16 (*m*/*z* [M + Na]^+^ C_48_H_72_O_10_N_6_Na (calcd. 915.5202)), dolastatin 16 ([M + Na]^+^ C_47_H_70_O_10_N_6_Na (calcd. 901.5046)), and antanapeptin A (C_41_H_60_O_8_N_4_ (calcd. 759.4314)) (Figure 1).

The detection of molecular ions corresponding to homodolastatin 16 from *M. producens* in the current study was consistent with that of the HRFABMS *m*/*z* [M+Cs]^+^ 1025.4364 (calcd for C_48_H_73_N_6_O10Cs, 1025.4364) obtained earlier [21]; however, this work represents the first report of the detection of dolastatin 16 in the Kenyan *M. producens*. The dolastatins are a group of antineoplastic pseudopeptides initially isolated from the sea hare *Dolabela auricularia* but that originate from the marine cyanobacterium *M. producens* distributed pan-tropically worldwide. Debromoaplysiatoxin (DAT) that is associated with *M. producens*’ swimmers’ itch was not detected by LCMS in the Kenyan strain of *M. producens*. The non-axenic nature of the Kenyan *M. producens* has been reported earlier [19] and was consistent with the isolation of the filament bacteria *Pseudoalteromonas carrageenovora* and *Ochrobactrum anthropi* from detached *M. producens*’ filaments on marine agar 2216 (10% *w*/*v*).

### 2.2. Phylogenetic Divergence of M. producens from L. majuscula

*L. majuscula* CCAP 1446/4 strain from the Culture Collection, Oban, Scotland, United Kingdom, and 16S rDNA analysis were used to delineate the Kenyan *M. producens* from other *L. majuscula* species elsewhere. Treatment of the cyanobacteria with cycloheximide, several rinses with phosphate buffered saline (PBS), and nitrogen liquefaction ensured that detached filaments were free from eukaryotic cells, protozoa, and fungi. Further, the cyanobacteria were treated with copper sulphate powder (CuSO_4_·5H_2_O) and left to stand for 10 min, 30 min, and 60 min, respectively, to kill any available bacteria. The control was untreated. Pooled detached filaments of both *L. majuscula* CCAP 1446/4 strain and the Kenyan *M. producens* biomass, free from eukaryotic cells, protozoa, and fungi, respectively, but with dead bacteria afforded sufficient biomass for gDNA extraction and genome sequencing. Crushing and homogenization of the biomass using a bench top bead-based homogenizer (PowerLyzer^TM^, 5 min) (MO BIO LABORATORIES Inc., California, USA) produced a homogenate from which bacterial gDNA was exhaustively extracted and the residue retained. CuSO_4_·5H_2_O assisted differential bacterial gDNA isolation of the residue of pre-extracted non-axenic filamentous cyanobacteria biomass followed by nitrogen liquefaction and sonication generated quality gDNA of *M. producens* for genome sequencing. The gDNA isolated from the residue provided high-purity DNA with 260/280 and 260/230 ratios of between 1.90 and 2.29 in the qubit assay, respectively. Treatment of the cyanobacteria biomass for times greater than 60 min denatured the *M. producens* DNA (Appendix A). The untreated *M. producens* control did not generate any pure DNA. Sequences of the Kenyan *M. producens* at the aforementioned times of 10 min, 30 min, and 60 min were synonymous with Kenyan *Moorea producens* Rep 1, Kenyan *Moorea producens* Rep 2, and Kenyan *Moorea producens* Rep 3, respectively, shown on the phylogeny diagram (Figure 2).

Initial NCBI blasts matched the cyanobacterium with *Aminobacterium colombiense*. However, there was a distinct difference in the sizes of DNA for bacteria and cyanobacteria observable in an electrophoresis gel. Moreover, mismatches arising from the primer CYA 106F were not unusual [29]. To validate the method, it was used to identify *L. majuscula* CCAP 1446/4. Further NCBI blasts of the generated sequence of the Kenyan *M. producens*, without restricting organism identity, matched the sequence at 99% identity with an uncultured *Aminanaerobia bacterium*. With the blast restriction to cyanobacteria, the sequences matched with an uncultured cyanobacterium at 98% identity. The 16S rDNA sequences obtained from *L. majuscula* strain CCAP 1446/4 by the aforesaid method for five replicates did not show any matches to *Aminanaerobia bacteria* but consistently matched with *L. majuscula* at 100% identity. The phylogenetic divergence of Kenya *M. producens* from *L. majuscula* and its clones is shown in Figure 2 and the blasted sequences are shown in Appendix A.

### 2.3. Bacterial Isolates from M. producens’ Filaments

In the quest to establish if *M. producens*’ filaments could have their associated bacteria completely removed, detached filaments were stained with acridine orange to establish cyanobacteria-bacteria association of the filament and with nigrosin to study cell wall and DNA degradation of the filament by bacteria.

Clearly, despite treatment of the cyanobacteria with cycloheximide, several rinses with phosphate buffered saline (PBS) and nitrogen liquefaction, microscopy and the isolation of the live filament bacteria (LFB) and dead filament bacteria (DFB) from the filament surface indicated that bacteria are always associated with *M. producens* (Figure 3A). Treatment of the cyanobacteria biomass with CuS0_4_·5H_2_O killed all bacteria except *Pseudoalteromonas carrageenovora* and *Ochrobactrum anthropi*. These surviving bacteria associated with the filament were plated on marine agar and identified through 16S rDNA gene partial sequence. The association of bacteria with cyanobacteria was regardless of the cyanobacteria getting actively involved with phototropism or on the verge of dying. Oxygen is acknowledged to be poisonous towards cyanobacteria including *M. producens*. Bacteria utilize oxygen for respiration and cyanobacteria capture carbon dioxide during photosynthesis, creating an energy balance on the cyanobacteria–bacteria interface. This close metabolic dependency may explain the close physical association of bacteria with the protective sheath surrounding the *M. producens*’ filaments. When bacteria on the surface of an untreated filament were left overnight to die, it was observed that they did not enter the core of the filament targeting the DNA material. Nevertheless, there was considerable loss of cell wall material (Figure 3B), leading to the speculation that some of the bacteria on the surface survive on organic carbon from the cyanobacteria. These findings overnight were corroborated by a broken cell wall, as observed on a nigrosine-stained filament similarly left to die.

Most bacteria isolated from the filaments of *M. producens* are close relatives of human pathogenic bacteria. The presence of Firmicutes and Actinobacteria on the filaments that are ubiquitous with antimicrobial agents suggested that within the consortium some bacterial species, in addition to other roles, may provide chemical defense molecules, which benefit the bacterial cells as well as the host cyanobacterial cells. However, it was unclear how the living filament retained its cylindrical shape, despite the presence of many species of bacteria, some of which were cellulose degraders, as exhibited by the bacteria on a dying filament (Figure 4B).

Further, microscopy of the Kenyan *M. producens* also showed that this cyanobacterium is associated with stalked diatoms (Figure 5). The apparent presence of stalked diatoms on the *M. producens*’ filaments indicated that microbes from estuarine and freshwater habitats may have colonized their surfaces [30]. Stalked diatoms, in particular, are found in marshes, rivers, and lagoons and are characteristically in freshwater. In the present study, *M. producens*’ samples were taken from an area around the Shimoni creek. This creek is a freshwater river inlet into Shimoni Bay.

### 2.4. Isolation of Bacteria from M. producens and Their Antimicrobial Activities

#### 2.4.1. Isolation and Identification of Bacteria from *M. producens*’ Sheath

The collection of *Moorea producens*’ biomass at Shimoni (4°38′55″ S, 39°22′35″ E) and Wasini (4°39′18″ S, 39°22′14″ E), Kenya, at low tide was reported previously [19]. Bacterial isolates from the Kenyan *M. producens*’ biomass were streaked directly onto marine agar 2216 (10% *w*/*v*). This consistently led to the isolation of colored colonies that were re-streaked to obtain pure strains. The concentration of marine broth was altered between 1% and 10% to differentiate between bacteria growing under poor and rich nutrient conditions, respectively. *Pseudomonas stutzeri* (yellow) was isolated from the sub-culturing of colonies growing together with colonies of *Enterobacter cloacae* (creamy), a mixture that was characterized by a green pigment. There were diverse morphologies of bacterial isolates after overnight incubation including tiny colonies of *Bacillus subtilis*, glassy *Pseudomonas putida*, large soft *Shewanella algae* (purple), and *Pseudomonas stutzeri* (yellow), respectively. Hard colonies were observed, which were identified as *Bacillus licheniformis* (red). A list of representative epibiotic bacterial isolates from the Kenyan *M. producens* is shown in Table 1.

Nearly 70% of all isolates were γ-proteobacteria. Firmicutes were isolated in reasonable quantities (17%), whereas α-proteobacteria and Actinobacteria were minimal. It is possible that the cyanobacterium appropriated pathogenic bacteria of terrestrial origin at low tide. Direct streaking led to the isolation of 137 bacterial strains, from which 24 strains were selected for further study and their organic extracts examined for antimicrobial activities. In common with the *M. producens*’ filament, most bacteria isolated from the sheath were found to be close relatives of human pathogenic bacteria.

#### 2.4.2. Isolation of Brightly Colored Bacteria from *M. producens*’ Sheath during Neap Tide

*Moorea producens* was collected from the sampling sites at a time of minimal tide when the sun and the moon were in quadrature. The growing of sheath bacteria onto marine agar by swab cultures showed the Gram-negative *Pseudoalteromonas* sp. to be the dominant genus. Pseudoalteromonas genus belongs to the sulfur-oxidizing symbiont relatives of γ-*Proteobacteria* commonly associated with sea water. However, this dominance is prevailed by treatment of the *M. producens*’ sheath with 70% ethanol (*v*/*v*, 2 min) prior to swab culturing, thereby enhancing the growth of brightly colored colonies of bacteria. The 16S rDNA sequencing identified these brightly colored bacteria as *Nocardia cornyebacterioides* (red), *Paracoccus marcusii* (orange), *Micrococcus* sp. (yellow), *Stappia* sp. (pink), and *Bacillus* sp. BacB3, respectively. Additionally, small colonies of unidentified, creamy, Gram-negative rods were isolated. *Nocardia cornyebacterioides* belongs to the genus *Nocardiopsis*. Prior to 16S rDNA identification techniques, the genus *Nocardiopsis* was defined on the basis of chemotaxonomic markers [31] with phospholipids type III and the phospholipids phosphatidylcholine and phosphatidylmethylethanolamine as salient chemotaxonomic features [31,32]. Previously, *Nocardia cornyebacterioides* was reported as an unpublished strain SAFR-015 with accession number AY167850.1 gi:27497670 in the MWG 16S rDNA data-bank. *Paracoccus marcusii* belongs to the α-3 subclass of the *Proteobacteria* [33], whereas the genus *Stappia* belongs to the ecologically important carboxydotrophic bacteria.

Interestingly, the study established that *Pseudoalteromonas* sp. was inhibited by ethanol (70% *v*/*v*) only during the initial 15 min, after which the bacteria population increased linearly with time. This was against the common belief that 70% ethanol has strong sterilizing activity against common bacteria. The natural microbial environment of bacteria on the surface of *M. producens* was mimicked by growing various strains of bacteria in filter discs (6 mm) over a lawn of the dominant *Pseudoalteromonas* sp. Subsequently, these bacterial antagonism experiments showed *Micrococcus* sp. (yellow) and *Stappia* sp. (pink) to antagonize *Pseudoalteromonas* sp. by slowing the growth of the latter bacteria by a zone of 12 mm to 19 mm and of 6 mm to 8 mm, respectively. The diversity of bacteria associated with the cyanobacterium was higher at low tide and lower during neap tide.

#### 2.4.3. Antimicrobial Activities of Organic Extracts of Bacterial Isolates

Using bacteria isolated from the cyanobacteria, organic extracts of Firmicutes were screened against bacteria, fungi, and known pathogens. Organic extracts were comprised of acetone and methanol, whereupon the acetone extracts were clearly the most active and were, therefore, used for the screens. Aliquots of the extracts and a streptomycin control in methanol (1 mg/mL, 20 µL) were tested against Gram-negative *Escherichia coli* strains JW0451-2 and BW25113, respectively, and Gram-positive *Bacillus subtilis* DSM 10 and *Micrococcus luteus* 1790 in 96-well microtiter plates. The absorbances of the organic extracts of microbial isolates were determined by serial dilution in EBS medium (0.5% casein peptone, 0.5% protease peptone, 0.1% meat extract, 0.1% yeast extract, pH 7.0) for bacteria and MYC medium (1.0% glucose, 1.0% phytone peptone, 50 mM HEPES [11.9 g/L], pH 7.0) for yeast and fungi, and were cultivated (30 °C, 160 rpm, 24−48 h) with cell concentration adjustment to OD_600_ 0.01 for bacteria and OD_548_ 0.1 for yeast and fungi, respectively. Then, they were matched with absorbances of Streptomycin and Nystatin in IC_50_ determinations to afford the levels of inhibitions (antibacterial activities), as shown in Table 2. The antibacterial activities were ranked using a percentage scale with the highest activity being assigned a value of 100% (Table 2).

The % scale activity values were obtained as follows:$$\frac{\mathrm{Absorbance}\;\mathrm{of}\;\mathrm{microbial}\;\mathrm{organic}\;\mathrm{extract}}{\mathrm{Absorbance}\;\mathrm{of}\;\mathrm{control}\;\mathrm{antibiotic}\;\mathrm{at}\;1\;\mathrm{mg}/\mathrm{mL}}\times 100\%$$

Extracts of *B. marisflavi* JC556 (LS974830.1) were active against *Micrococcus luteus* and *B. subtilis*. Those of *Bacillus licheniformis* PB3 (CP025226.1) displayed moderate activity against *Escherichia coli* and *B. subtilis*, whereas organic extracts of *B. subtilis* MJ01 showed very high activity against *E. coli* and *M. luteus*, respectively. Biological activity was strain dependent, as demonstrated by *B. marisflavi* LQ1 (MG025780.1) that was less active compared to *B. marisflavi* JC556 (LS974830.1). Similarly, *Bacillus subtilis* TBS-CBE-BS01 (MK346244.1) had minor activity against *M. luteus* compared with *B. subtilis* MJ01. Organic extracts of the bacterial isolates showed no biological activities against *Pichia anomala* and *Mucor hiemalis*, and neither were biological activities of the extracts observed against the pathogens *Acinetobacter baumannii*, *Citrobacter freundii*, *S. aureus* Newman, *Pseudomonas aeruginosa* PA14, *Mycobacterium smegmatis,* and the yeast *Candida albicans*. Of the 24 strains, 5 that were considered not interesting due to potential pathogenesis were later dropped and are, therefore, not included in Table 2 above. Organic extracts of *B. marisflavi* JC556 (LS974830.1) and *B. subtilis* MJ01 were the most active. *B. marisflavi* extract was selected for bioassay-guided isolation of its natural products. Streptomycin was the reference antibiotic for bacteria and nystatin was a reference against yeast and fungi, respectively.

### 2.5. Isolation of Micrococcin P1 and Micrococcin P2 and Biological Activities

#### 2.5.1. Isolation of Micrococcin P1 (**1**) and Micrococcin P2 (**2**)

In this study, extracts of a number of cultured Firmicutes were found to inhibit the growth of *Bacillus subtilis* except for those from *B. subtilis*, suggesting that bacteria species can produce antibiotics against pathogens from their own genus. This observation could pave the way for exploring future antibiotics against *Pseudomonas aeruginosa* and *Klebsiella pneumoniae* that are responsible for nosocomial infections from members of the *Pseudomonas* genus and *Klebsiella* spp., respectively. Acetone extracts of *B. marisflavi* cultures were the most biologically active, especially against Gram-positive *Staphylococcus aureus*. The antibacterial compounds identified in these extracts were micrococcin p1 (**1**) and micrococcin p2 (**2**), shown below in Figure 6.

C-18 HPLC fractionation of the active fraction of *B. marisflavi* using maXis 2 G (BRUKER DALTONICS, Bremen, Germany) afforded the biologically active antibiotics micrococcin P1 (**1**) and micrococcin P2 (**2**) with the retention times of 9.43 min and 9.67 min, respectively (Appendix A). Using HREIMS, the LCMS-MS fragmentation with maXis 4G (BRUKER DALTONICS, Bremen, Germany) unequivocally isolated the known antibiotics micrococcin P1 and micrococcin P2 from *B. marisflavis* (Appendix A). Molecular formulae were identified by including the isotopic pattern in the calculation (SmartFormula algorithm). Both the fractionation with maXis 2G and the high-resolution MS/MS fragmentation using maXis 4G used an acetonitrile/water gradient system that was buffered with ammonium acetate. Details on how the results were acquired can be found in Section 3.7 of the experimental section.

Dereplication of the active molecular ions of 571.60/1142.2 and 572.60/1144.2 using MS^2^ ions in maXis 4G and the Dictionary of Natural Products linked the strong activity of the *Bacillus marisflavi* strain to its production of micrococcin P1 (**1**) (C_48_H_49_N_13_O_9_S_6;_ obs. *m*/*z* 1144.20227/572.60381) and micrococcin P2 (**2**) (C_48_H_47_N_13_O_9_S_6_; obs. *m*/*z* 1142.20446/571.60370), respectively. These antibiotics were originally isolated from *Micrococcus* sp. [34]. Their structures were confirmed by spectroscopy in 1978 [35] and the total synthesis and stereochemical assignment was accomplished in 2009 [36]. More recent work includes the ribosomal production of micrococcin P1 and micrococcin P2 antibiotics through prepeptide gene replacement [37]. Most recently, the antibiotic thiopeptide micrococcin P3 was isolated from a marine-derived strain of *Bacillus stratosphericus* [38]. In this study, we established the following MS-MS fragmentation pattern for micrococcin P1 (**1**) from the maXis 4G measurements (Table 3) that unequivocally matched with that reported by Walsh et al., in 2010 [37].

Micrococcin P1 and micrococcin P2 are closely related molecules with the former existing as the secondary alcohol and the latter as the ketone congener. The MS/MS fragmentation pattern shown above for micrococcin P1 is the most common pathway followed by thiocillin variants including micrococcin P2 [37]. It could, therefore, be concluded that together with high-resolution mass spectrometry, dereplication of the active molecular ions, and the Dictionary of Natural Products, bioassay-guided fractionation of the extracts of *B. marisflavi* led to the first isolation of the antibiotics micrococcin P1 (**1**) and micrococcin P2 (**2**) from a strain of *B. marisflavi*. Isolation of these known antibiotics was strain dependent since organic extracts of other strains of *B. marisflavi* isolates did not exhibit the activity of *Bacillus marisflavi* JC556 (LS974830.1).

#### 2.5.2. Biological Activity of Micrococcin P1 against *Staphylococcus aureus*

The activity of micrococcin P1 (**1**) against a panel of pathogens, *Acinetobacter baumannii*, *Citrobacter freundii*, *S. aureus* Newman, *Pseudomonas aeruginosa* PA14, *Mycobacterium smegmatis*, and the yeast *Candida albicans*, was minimal with activity only observed against *Staphylococcus aureus*. This last activity of micrococcin P1 against *S. aureus* was consistent with that in the literature [33,34,39]. MIC is the lowest concentration of the drug, preventing visible growth of the test organism [40], and IC_50_ is the concentration of the drug at which the growth of half of the test organism (bacteria, fungi or yeast) is inhibited [41]. An MIC value of 0.0157 µM was obtained for micrococcin P1. Using the method of Okanya et al. [41] and its modification as outlined by Jansen et al. [40] and Bader et al. [42], the IC_50_ of micrococcin P1 against *S. aureus* was shown to be 0.357 µM (0.159 mg/mL) (Appendix A). This value is within the range of IC_50_ of micrococcin P1 against an assorted collection of *S. aureus* strains and confirms the activity of micrococcin P1 against *S. aureus* to be strain dependent [39]. The IC_50_ regression curve of the biological activity data of micrococcin P1 against *S. aureus* is shown in Appendix A.

## 3. Experimental

### 3.1. Study Area and Specimen Preparation

*Moorea producens*’ specimens were collected from Shimoni (4°38′55″ S, 39°22′35″ E) and Wasini (4°39′18″ S, 39°22′14″ E) on the South Coast of Kenya during 2012 (during neap and low tides) and 2018 (low tide). Specimens were preserved in sterile PTFE bottles prior to storage at −20 °C, prior to shipment to the UK (2012) and Germany (2018). Bacteria were isolated from *M. producens* using traditional methods and identified using Gram stain and microscopy in Kenya. At Newcastle University, United Kingdom, and at the Helmholtz Institute, Germany, the bacterial isolates were identified using microscopy and 16S rDNA identification.

### 3.2. Detection of Homodolastatin 16, Dolastatin 16, and Antanapeptin A in Kenyan Isolates of a Filamentous Cyanobacterium, Moorea producens

The detection of homodolastatin 16, dolastatin 16, and antanapeptin A in *M. producens* was confirmed by LCMS analysis of organic extracts of the cyanobacterium. Briefly, *M. producens* (previously known as *L. majuscula*) (2 g) was extracted with 2:1 dichloromethane/methanol (50 mL, 12 h, 25 °C, sonicated) and filtered through glass fiber filters (44 μ). Fractionation of the extracts from the Kenyan “*L. majuscula*” was achieved on C-18 silica to give “TD-Lyng chl” (the first Lyngbya extract fraction to elute from the C-18 column) and “TD-Lyngbia” that eluted immediately after “TD-Lyng chl”. TD-Lyng chl and TD-Lyngbia were respectively evaporated and dried under nitrogen. The dried extracts were redissolved in 1:1 methanol/water to a final concentration of 2 mg/mL. Subsequently, 5 μL of the sample was injected onto the column (XBridge C18, 3.5 μm, 2.1 × 150 mm) and separated on a gradient of 10% 0.1% formic acid in water/90% 0.1% formic acid in acetonitrile to 5% 0.1% formic acid in water/95% formic acid in acetonitrile at 0.3 mL/min. Molecular ions for the cyclodepsipeptides were identified by Fourier transform mass spectrometry (FTMS) from ESI Full MS chromatograms. The MS used was a Thermo Scientific Orbitrap Exactive (THERMO FISHER SCIENTIFIC, Bremen GmbH, Germany); resolution was set to 150,000.

### 3.3. CuSO_4_·5H_2_O Assisted Differential Genomic DNA (gDNA) Extraction of M. producens

Prior to bacteria genomic DNA (gDNA) isolation, the cyanobacteria cells were treated with cycloheximide (5 mg L^−1^, overnight) to kill eukaryotic cells, protozoa, and fungi. Thereafter, the cells were rinsed several times with filtered sterile seawater (12 times, 45 μ) and left overnight in phosphate-poor autoclaved seawater [29]. For detaching filaments, clusters of Moorea biomass were submerged in phosphate buffered saline (PBS, pH 7.4). The resulting filaments were rinsed with Millipore water and pooled together, and aliquots were weighed to afford sufficient biomass for gDNA extraction and genome sequencing. Thereafter, the cyanobacteria were treated with copper sulphate powder (CuSO_4_·5H_2_O) and left to stand for 10 min, 30 min, and 60 min, respectively, to kill any available bacteria. The control was untreated. For removing the bacteria, biomass of pooled filaments was placed in an Eppendorf tube, freeze-dried with liquid nitrogen, and thawed slowly. The tube contents were sonicated (pulsar 30% of maximal amplitude, 10 min) and the process was repeated 5 times. Representative filaments were stained with acridine to observe the presence of any associated bacteria using a LEICA DMIRB microscope (LEICA DMIRB, Mannheim, Germany) with Media Cybernetics image analysis software Image-Pro Plus 7 (MEDIA CYBERNETICS, Rockville, MA, USA).

Aliquots of the biomass (0.25 g) were weighed in sterile MoBio microfuge tubes into which was added 100 μL of sodium dodecyl sulfate (SDS) and 0.5 mL of RNAse free TE buffer (50 mM Tris HCl, 20 mM EDTA, pH 8.0). The specimen was crushed and homogenized using a bench-top bead-based homogenizer (PowerLyzer™, 5 min). The resulting homogenate was transferred into an Eppendorf tube, freeze-dried in liquid nitrogen, and thawed slowly. The tube contents were sonicated (pulsar 30% maximal amplitude, 10 min) and the process was repeated five times [43]. The sample was centrifuged (10,000× *g*, 3 min) and bacterial gDNA were extracted following the MoBio kit for soil bacteria; the residue was retained. The DNeasy PowerMax Soil Kit used here was formerly sold by MOBIO as PowerMax Soil DNA Isolation Kit, and this is currently sold by Qiagen (QIAGEN, Valencia, CA, USA). The residue was re-extracted for bacterial gDNA until no traces of the gDNA were detected on an electrophoresis gel.

To the bacteria-free residue was added 1 mL of Lysis buffer (TE buffer with RNase) and heated (55 °C, 1 h) prior to adding 300 μL of freshly prepared lysozyme at 50 mg/mL. The mixture was heated (55 °C, 30 min). Into the contents was added 100 μL of a solution containing 450 μL of 10% sarcosyl and 20 μL of 20 μg/mL proteinase k and incubated (55 °C, 30 min). To each of the contents in a 2-mL Eppendorf tube was added 3 volumes of phenol: chloroform: isoamyl alcohol (25:24:1) and the contents of the tubes were mixed with inversion and centrifuged (10,000× *g*, 5 min). The resulting gDNA was isolated and purified following conventional standard methods [44]. Purity and integrity of the gDNA was confirmed using a combination of spectrophotometry (NanoDrop) and agarose gel electrophoresis [44].

PCR amplification of the purified gDNA was done according to the protocol of Nubel et al. [29]. The forward primer is CYA106F (5′-CGGACGGGTGAGTAACGCGTGA-3′), whereas the reverse primer is an equimolar mixture of CYA781Ra (5′-GACTACTGGGGTATCTAATCCCATT-3′) and CYA781Rb (5′-GACTACAGGGGTATCTAATCCCTTT-3′) [29]. Preparatory columns (Invitrogen PCR Preps) were used to clean the PCR products. PCR products were sequenced by Genius labs, UK, and the sequences were annotated using Bioedit 7. Alignment of the sequences and phylogeny of the aligned sequences utilized the MEGA X program [45]. For obtaining gDNA of the cyanobacterium, homogenized *L. majuscula* pellets were exhaustively extracted for bacteria gDNA. The resulting bacterial gDNA of mixed species was degenerated by CuSO_4_·5H_2_O and therefore it did not generate a 16S rDNA sequence.

### 3.4. Isolation of Bacteria from M. producens’ Filaments

Preparation of *M. producens* to detach filaments and to determine its (*M. producens*) gDNA using CuSO_4_·5H_2_O is described in Section 3.3. In another experiment, filaments from cyanobacteria biomass that was treated with CuSO_4_·H_2_O were plated onto marine agar to determine the assemblage of bacteria associated with the filaments on a cyanobacterial sheath. These experiments were repeated for a near-dead *M. producens* mat. Bacteria isolated from the live and dead filaments were designated as LFB and DFB, respectively, prior. These bacterial isolates were identified using 16S rDNA partial gene sequence. Additional filaments from the live and near-dead *M. producens* were thoroughly washed and stained with acridine orange and nigrosin to establish association of the filaments with heterotrophic bacteria.

### 3.5. The 16S rDNA Identification of Bacterial Isolates

Genomic DNA was extracted from cell pellets of single colony bacterial cultures in marine broth 2216. The Qiagen DNeasy Blood and Tissue kit was used (QIAGEN, Valencia, CA, USA). Specifically, overnight culture containing approximately 1–3 × 10^9^ bacterial cells were transferred to a 1.5-mL Eppendorf tube on ice and centrifuged (13,000× *g*, 1 min). Prior to the addition of RNase buffer (1.5 µL), the cells were lysed with lysis buffer (300 µL, 65 °C, 15 min), incubated (37 °C, 15 min), and centrifuged. The resulting supernatant from centrifugation (13,000× *g*, 1 min) was thereafter treated with genomic-binding buffer (0.5 mL), centrifuged (10,000 rpm, 2 min), and passed through a DNA-binding column. Washing the column with genomic wash buffer (1 mL) and three times with 1 mL of 75% ethanol followed by dry spinning afforded purified DNA on the column. The resulting DNA was eluted from the column with milliQ water.

Reacting a mixture containing 10 × Qiagen buffer (2.5 μL), MgCl_2_ (0.5 μL), d’NTPS (1 μL), 2.5 μL of primer 27 F (5′-AGAGTTTGATCCTGGCTCAG-3′), and 2.5 μL of primer 1492 R (5′-TACGGYTACCTTGTTACGACTT-3′), DMSO (0.75 μL) and Taq polymerase (0.1 μL) gave the polymerase chain reaction (PCR) product of the 16S rDNA. Sterile Milli Q water was added to adjust the final volume of the PCR mixture to 25 μL. Thermal cycling was performed with an Eppendorf Mastercycler AG 22331 (EPPENDORF, Hamburg, Germany). The sample utilized the standard initial denaturation step (95 °C, 5 min; 95 °C, 30 s) followed by annealing (40 °C, 1 min). Procedurally, the thermal profile used was 30 cycles. This profile consisted of 1 min of primer annealing at 55 °C, 1 min of extension at 72 °C, and 1 min of denaturation at 95 °C. A final extension step consisting of 10 min at 72 °C was also included. PCR products were detected by agarose gel electrophoresis and visualized by UV fluorescence after ethidium bromide staining. The PCR products generated by the 27 F and 1492 R primers were approximately 1 kb in size and purified using the Qiagen genomic DNA as specified by the manufacturer. Sequencing of the final DNA extracts was done by LGC Genomics (Germany).

### 3.6. Antibacterial and Antifungal Assays

Fifty bacterial strains out of 137 were selected on the basis of color and morphological appearance, from which 24 were sequenced and their respective 16S rDNA sequences were identified by BLAST (BLASTN 2.8.1+) [46]. Cultures were maintained in marine broth, and pre-cultures were grown on medium 5294 [47] in a rotatory shaker incubator (50 mL, 30 °C, XAD resin, 3 days). Precultures were in duplicate. Cultures of the bacteria for secondary metabolite production were grown in Medium 5294 (2% XAD resin, 7 days, 37 °C). The XAD resin was used to sequester secondary metabolites from the media. Specifically, the resin was initially extracted with acetone prior to extraction with methanol. The acetone extracts, being more active than the extracts of methanol, were reconstituted in methanol. Aliquots of the acetone extracts in methanol were used for bioassays against target microbial screens and mass spectrometry profiling. Mass spectrometry profiling is invaluable in this context as it provides all the available masses of the extract.

The antibacterial and antifungal assays adopted the method of Jansen and co-workers [40], briefly described below. Aliquots of the extracts, reference antibiotics, and micrococcin P1 (**1**) (1 mg/mL, 20 µL) were tested against Gram-negative *Escherichia coli* strains JW0451-2 and BW25113, respectively, and Gram-positive *Bacillus subtilis* DSM 10 and *Micrococcus luteus* 1790. Antifungal screens utilized *Pichia anomala* and *Mucor hiemalis*. Streptomycin (SIGMA-ALDRICH) was the reference antibiotic for bacteria and nystatin (SIGMA-ALDRICH) was a reference against yeast and fungi, respectively. *Acinetobacter baumannii*, *Citrobacter freundii*, *S.aureus* Newman, *Pseudomonas aeruginosa* PA14, *Mycobacterium smegmatis*, and the yeast *Candida albicans* were used for assays against pathogens. The % scale activity values were determined in 96-well microtiter plates by 1:1 serial dilution in EBS medium (0.5% casein peptone, 0.5% protease peptone, 0.1% meat extract, 0.1% yeast extract, pH 7.0) for bacteria and MYC medium (1.0% glucose, 1.0% phytone peptone, 50 mM HEPES [11.9 g/L], pH 7.0) for yeasts and fungi. The test organisms were cultivated at 30 °C and 160 rpm for 24−48 h, and the cell concentration was adjusted to OD_600_ 0.01 for bacteria and OD_548_ 0.1 for yeast and fungi before the test. The antibacterial activities were ranked to % scale (Table 2). The activities of micrococcin P1 (**1**) against *S. aureus* were given an activity score relative to their absorbances and matched with their respective concentrations. The lowest concentration of micrococcin P1 inhibiting the growth of *Staphylococcus aureus* was considered to be the MIC. The IC_50_ of micrococcin P1 against *S. aureus* used the method in Okanya et al. [41]. An IC_50_ calculator was used to generate the IC_50_ regression curve and to obtain the IC_50_ of micrococcin P1 (**1**) against *S. aureus*.

### 3.7. Isolation of Micrococcin P1 (**1**) and Micrococcin P2 (**2**)

The active fraction of *B. marisflavi* JC556 (LS974830.1) was cultivated overnight and fractionated according to the respective OD_600_ (0.01 for bacteria and OD_548_ 0.1 for yeast and fungi) using maXis 2 G (BRUKER DALTONICS, Germany); the retention times of the entire 96-well plate were recorded. Fractions with the lowest absorbances were considered to be the most active. LCMS-MS fragmentation of the compounds with the retention times of 9.43 min and 9.67 min, respectively, was realized with maXis 4G (Bruker, Germany). Generally, the methods of Okanya et al. were adopted [41]. Accordingly, HREIMS data were recorded on a maXis spectrometer (BRUKER DALTONICS, Germany). Molecular formulae were identified by including the isotopic pattern in the calculation (SmartFormula algorithm). Analytical RP HPLC utilised an Agilent 1100 HPLC system (AGILENT TECHNOLOGIES, Waldbronn, Germany) equipped with a UV diode-array detector and an evaporative light-scattering (PL-ELS 1000) detector. HPLC conditions were column 125 × 2 mm, Nucleodur C-18, and 5 μm (MACHEREY-NAGEL, Düren, Germany); solvent A, 5% acetonitrile in water, 5 mmol of NH_4_Ac, and 0.04 mL of CH_3_COOH; solvent B, 95% acetonitrile, 5 mmol of NH_4_Ac, and 0.04 mL of CH_3_COOH; gradient system, 10% B increasing to 100% in 30 min, 100% B for 10 min, to 10% B post-run for 10 min; 40 °C; and flow rate of 0.3 mL/min. Dereplication of these compounds along with the Dictionary of Natural products were used to identify the known antibiotics micrococcin P1 (**1**) and micrococcin P2 (**2**) from a strain of *B. marisflavi*.

## 4. Conclusions

The Kenyan isolate of “*Lyngbya majuscula*”, also known as *Moorea producens* and recently renamed Moorena, was shown to be phylogenetically distinct from *L. majuscula* CCAP 1446/4 and *L. majuscula* clones. Surprisingly, epibiotic bacteria growing on the surface of *M. producens*’ filaments were often found to be human pathogens. A new method for the isolation of genomic DNA (gDNA) from non-axenic filamentous marine cyanobacteria based on the CuSO_4_·5H_2_O assisted differential isolation of bacteria gDNA from a cyanobacteria biomass was reported. Debromoaplysiatoxin (DAT), which is an indicator toxin found earlier in *M. producens*, was not detected in the Kenyan strain of *M. producens*. The phylogenetic divergence, absence of DAT, and the detection of unique cyclodepsipeptides in the Kenyan strain of *M. producens* reinforce a need for molecular identification of non-axenic *M. producens* species worldwide. The IC_50_ of micrococcin P1 against *S. aureus* was within the range of values reported in literature. Our results showed that the genetic basis for synthesizing micrococcin P1 (**1**), originally discovered in *Bacillus cereus* ATCC 14579, is often species or strain dependent and provides evidence for *M. producens*-associated bacteria as an underexplored source of bacterial strains for the discovery of new antibiotics to fight drug resistance.

## Figures and Tables

**Figure 1 marinedrugs-20-00128-f001:**
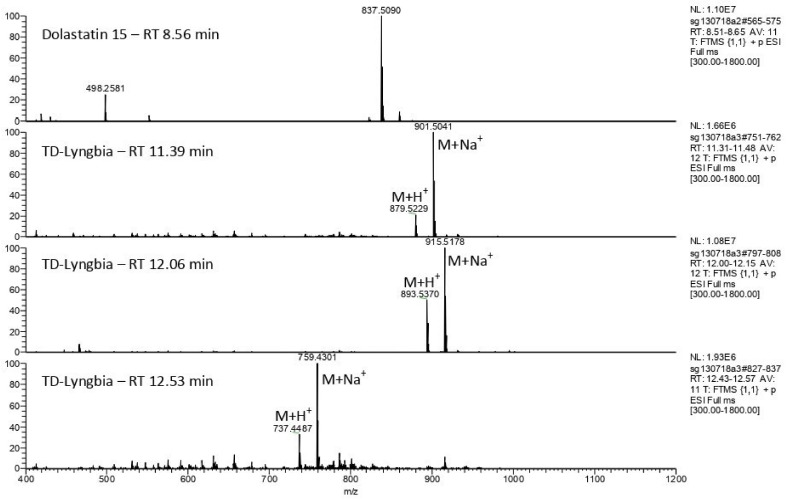
Electrospray mass spectrometry showing molecular ions of dolastatin 16, homodolastatin 16, and antanapeptin A from LCMS analysis of Kenyan *M. producens*’ extracts at retention times of 11.39 min, 12.06 min, and 12.53 min, respectively. Note: dolastatin 15 included as standard. LCMS conditions: gradient of 10% 0.1% formic acid in water/90% 0.1% formic acid in acetonitrile to 5% 0.1% formic acid in water/95% formic acid in acetonitrile at 0.3 mL min^−1^.

**Figure 2 marinedrugs-20-00128-f002:**
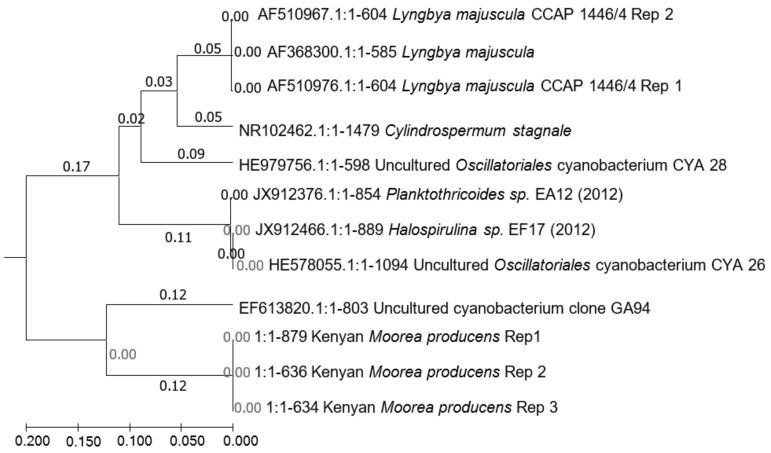
Phylogeny of Kenyan *M. producens* relative to *Lyngbya majuscula* and *Lyngbya majuscula* CCAP 1446. Both *L. majuscula* and *L. majuscula* CCAP 1446 are phylogenetically far distant from the Kenyan *M. producens*. Kenyan *M. producens* Rep 1 (CuSO_4_·5H_2_O, 10 min), Kenya *M. producens* Rep 2 (CuSO_4_·5H_2_O, 30 min), Kenyan *M. producens* Rep 3 (CuSO_4_·5H_2_O, 60 min). Rep 1 and Rep 2 of *L. majuscula* CCAP 1446 are sequence replicates of the biomass treated with CuSO_4_·5H_2_0 at 10 min.

**Figure 3 marinedrugs-20-00128-f003:**
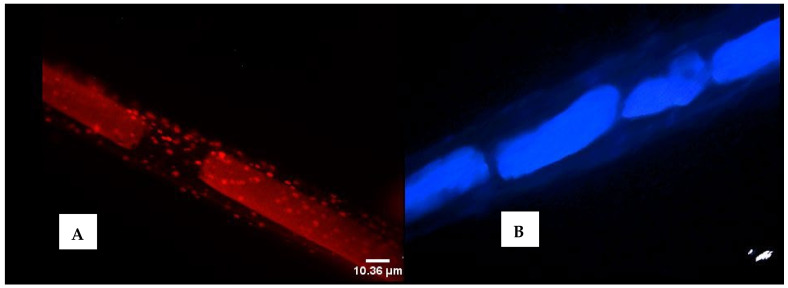
Bacteria on the surface of a live *M. producens*’ filament (**A**) and DNA in blue at the center and cell wall material outside (**B**). The images of the filaments on a # 1.5 (~170-µm-thick) glass cover slip were acquired by the author using a Leica DMIRB inverted microscope fitted with a color and black-and-white cooled digital camera. Acridine orange was used to stain the filament shown in (**A**) and nigrosin stained the filament shown in (**B**). Scale bar = 10.36 µm.

**Figure 4 marinedrugs-20-00128-f004:**
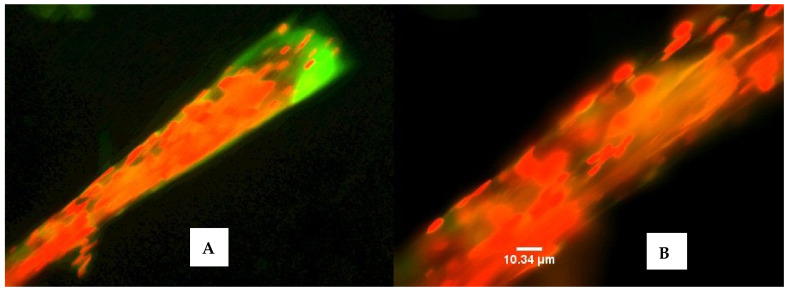
A non-uniform near-dead filament with congregation of live bacteria on the dead filament shown by the red color (**A**). Bacteria embedded in surface of a dying filament (**B**). The green and brown areas in (**A**,**B**) are the live and dead parts of the filament, respectively. The images were acquired by the author, as in Figure 3 above. Staining of the filament was with acridine orange. Scale bar = 10.34 µm.

**Figure 5 marinedrugs-20-00128-f005:**
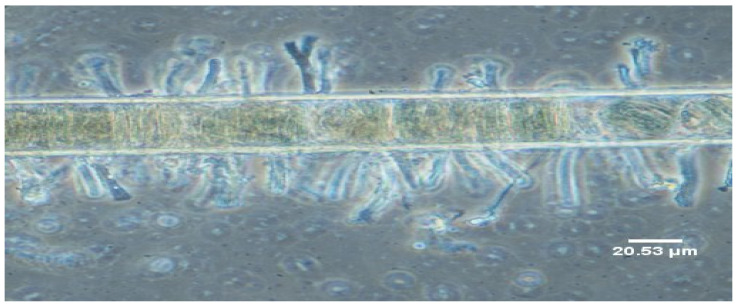
Filament of the Kenyan *M. producens* with stalked diatoms protruding from both its sides. Light microscopy of the filament with an OLYMPUS BX 51 equipped with inter-differential contrast optics and digital camera. The image was acquired by the author. Scale bar = 20.53 µm. Stalked diatoms were confirmed as being of a terrestrial nature by Prof. Brian Whitton of Durham University, United Kingdom.

**Figure 6 marinedrugs-20-00128-f006:**
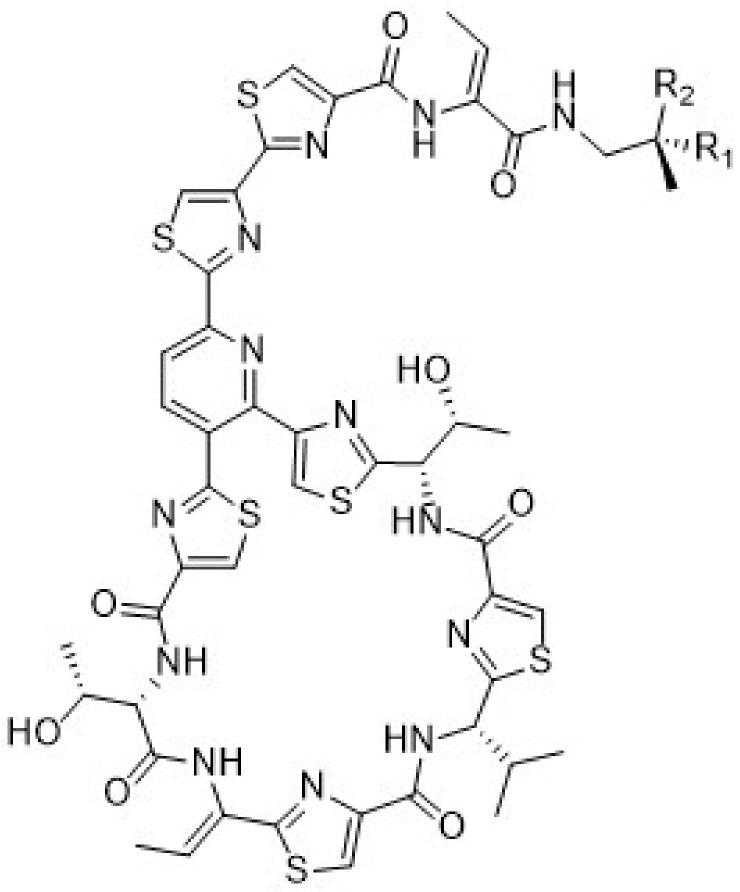
Micrococcin P1 (**1**): R_1_ = OH, R_2_ = H; Micrococcin P2 (**2**): R_1_ = R_2_ = O.

**Table 1 marinedrugs-20-00128-t001:** A list of representative Kenyan *M. producens* epibiotic bacteria (EB) isolates.

	Accession	Strain	% Similarity of 16S Sequence	Taxon
*Shewanella algae*	KC660130	SHALG-01	99	γ-proteobacteria
*Shewanella algae*	KC660131	SHALG-02	99	γ-proteobacteria
*Marinobacterium stanieri*	KC660132	MARIS-01	99	γ-proteobacteria
*Acinetobacter johnsonii*	KC660133	ACJ-01	99	γ-proteobacteria
*Marinobacterium stanieri*	KC660134	MARIS-02	99	γ-proteobacteria
*Staphylococcus saprophyticus*	KC660135	STAPRO	99	Firmicutes
*Pseudomonas stutzeri*	KC660136	PST-01	99	γ-proteobacteria
*Enterobacter cloacae*	KC660137	ENTCLO	99	γ-proteobacteria
*Cellulosimicrobium cellulans*	KC660138	CCL-01	99	Actinobacteria
*Cellulosimicrobium cellulans*	KC660139	CCL-02	99	Actinobacteria
*Pseudomonas pseudoalcaligenes*	KC660140	PPS	99	γ-proteobacteria
*Pseudomonas putida*	KC660141	PPT	99	γ-proteobacteria
*Bacillus aereus*	ND	ND	99	Firmicutes
*Bacillus licheniformis*	KC660142	BLC-01	99	Firmicutes
*Bacillus licheniformi*	KC660143	BLC-02	99	Firmicutes
*Bacillus subtilis*	KC660144	BS-00	99	Firmicutes
*Pseudomonas stutzeri*	KC660145	PST-02	99	γ-proteobacteria
*Enterobacter cancerogenus*	ND	ND	99	γ-proteobacteria
*Klebsiella oxytoca*	ND	ND	99	γ-proteobacteria
*Yokenella regensburgei*	ND	ND	99	γ-proteobacteria
*Ochrobactrum* *anthropi*	ND	ND	99	α-proteobacteria
*Pseudomonas stutzeri*	ND	ND	99	γ-proteobacteria
*Pseudoalteromonas carrageenovora*	ND	ND	99	γ-proteobacteria

ND–Strain sequences not deposited with Genbank but were inferred from Blast.

**Table 2 marinedrugs-20-00128-t002:** Antimicrobial activity of bacterial isolates from a marine cyanobacteria.

Bacterial Strains	Closest Match in GenBank (Accession Number)	% Similarity of 16S Sequence	Antimicrobial Activity	Level of Inhibition
TD1, TD15	*Bacillus marislavi* JC556 (LS974830.1)	100	*M. luteus*, *B. subtilis*	Active
TD2, TD26	*Bacillus marislavi* LQ1 (MG025780.1)	100	*B. subtilis*	Minor activity
TD3, TD6	*Bacillus safensis* 6-11 (MK205159.1)	100	*B. subtilis*	Minor activity
TD10	*Bacillus safensis* 6-11 (MK210556.1)	100	Not Done	
TD11	*Bacillus safensis* D11 (KX068630.1)	100	*B. subtilis*, *P. anomalis*	Minor activity
TD23, 25	*Bacillus safensis* D11 MK337676.1	100	*M. luteus*, *B. subtilis*	Minor activity
TD4	*Bacillus aryabhattai* PYMW (MK346120.1)	100	*B. subtilis*, *P. anomalis*	Minor activity
TD8	*Bacillus aryabhattai* P6 (MK346850.1)	100	*B. subtilis*	Minor activity
TD5, TD22	*Bacillus licheniformis* PB3 (CP025226.1)	99	*E. coli*, *B. subtilis*	Moderately active
TD7	*Bacillus subtilis* MJ01	100	*E. coli*, *M. luteus*	Very active
TD12	*Bacillus subtilis* TBS-CBE-BS01 (MK346244.1)	100	*M. luteus*	Minor activity
TD9	Sequence failed	Not done	*B. subtilis*	Minor activity
TD13	*Arthrobacter* sp. ABCH 95.B (KY327809.1)	100	NA	No activity
TD14	*Bacillus flexus* 00F26 MH542283.1	Not done	NA	No activity

KEY: 0–24%, No activity; 25–49%, Minor activity; 50–74%, Moderate activity; 75–99%, Active; 100%, Very active. Activities were relative to absorbances of Streptomycin and Nystatin in an IC_50_ determination.

**Table 3 marinedrugs-20-00128-t003:** Fragmentation pattern of micrococcin P1 (**1**) 1144.20227/572.60381.

	Molecular Ion	Expected *m*/*z* (a.m.u)	Observed *m*/*z* (a.m.u)
1	M + H	1144.21732	1144.21930
2	M − H_2_O + H	1126.20676	1126.20946
3	M − CO	1116.22241	1116.22442
4	M − 2H_2_O + H	1108.19619	1108.19865
5	M − CO − H_2_O	1098.21184	1098.21380
6	M − CO − 2H_2_O	1080.20128	1080.20333
7	b13 + H_2_O	1051.13834	1051.14040
8	a13	1041.15399	1041.15222
9	b13 − 2H_2_O	1033.12778	1033.12980
10	b2 + H_2_O	1025.15908	1025.16113
11	a13 − H_2_O	1023.14343	1023.14492
12	b2 − 2H_2_O	1007.14851	1007.15051
13	a13 − 2H_2_O	1005.13286	1005.13482

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
