# Peer review of "Micrococcin P1 and P2 from Epibiotic Bacteria Associated with Isolates of Moorea producens from Kenya"

_marinedrugs, 2022, doi:10.3390/md20020128_

Round 1

Reviewer 1 Report

The authors take into account the previous remarks. They isolated micrococcin P1 and P2 from their strain. They tried to perform MIC analysis but did not succeeded. They proposed the Table 2 and indicated a level of inhibition. It is just an indication of the level of inhibition. Perhaps the authors could present their results in a dot box plot because here is it difficult to read and summarized the results. It is also no very clear how the authors performed the experiments. Which extract is used against which microorganism? It will be necessary to perform quantitative analysis. The main information is diluted that it is Moorra producens which produces antibiotic.

Reviewer 2 Report

  1. One novelty of this study is identification of known natural products, micrococcins P1 and P2 in a new bacterial producer (B. marisflavi, living in symbiosis with Moorena producens). Micrococcins had been intensively studied over more than 20 years and isolated from various species. So, presenting their structure in Figure 6 is probably redundant. Also, the fact that their identification is solely based on mass spectrometry does not provide a convincing prove of their identity. An alternative technique, such as NMR or MS-MS would be more convincing as these natural products could be unknown isomers of the known compounds. 
  2. Protocol for microccocins isolation is added but poorly written. I believe the phrases "OD602 (0.01 for bacteria and OD548 0.1 for yeast and fungi)" on line 455 as well as "the retention times of the entire 96 well plate" on line 456 do not belong there. Moreover, the critical information, like mobile phase composition used, is not mentioned. 
  3.  The second novelty of the study is presentation of a new method for the isolation of genomic DNA from non-axenic cyanobacteria (lines 466-467). However, the success of this method is not well represented.
  4. The manuscript states (lines 105-108) that bacteria identity ( Moorea) could be based on the secondary metabolites it produces. To the best of my knowledge, this is not supported by the experimental evidence. 
  5. The literature background on Moorea/Moorena species is not sufficient. Also, genetic tree to prove Moorea/Moorena identity has limited number of referenced organisms.
  6. According to the 3.6-chapter MIC protocol was performed on the extracts but MIC values were not recorded for some unexplained reason, while ‘level of inhibition’ was reported instead
  7. The major achievement of this study is identification of a huge variety of symbiont organisms associated with the strain of filamentous cyanobacteria

Reviewer 3 Report

Can the authors provide the NMR and 2D data for micrococcin P1 and P2 nd for dolastatin?

Author Response

Please attachment

Round 2

Reviewer 1 Report

The authors try to explain how they calculate the antimicrobial activity. They gave a formula but in Table 2, there is no quantitative value. The results are expressed as very active or moderate/minor or no activity. If the authors have calculated the value of antimicrobial activity, it will be better to give the values and not to give qualitative data. 

It could be also interesting to reorganize the paper with sequential information, with first the phylogeny, second isolation of antimicrobial product and caracterization, third the antimicrobial activity caracterization and finally the microscopy study. It is not clear the activities measurements from bacterial isolates and finally the activity measurement on Staphylococcus aureus. Did the authors made the experiment "against a panel of pathogens Acinetobacter baumanii, Citrobacter freundii, S. aureus Newman, Pseudomonas aeruginosa PA14, Mycobacterium smeg matis and the yeast Candida albicans was decimal with only a marked activity against Staph ylococcus aureus."

A reorganization and clarification of the paper seems to be important. What is the main information that the authors want to give?

Reviewer 2 Report

I do not have any

Author Response

Reviewer 2 had no comments during round 2

This manuscript is a resubmission of an earlier submission. The following is a list of the peer review reports and author responses from that submission.

Round 1

Reviewer 1 Report

The authors describe the study for epibiotic bacteria associated with the filamentous marine cyanobacterium Moorea producens to identify

Line 212, Fig. 4 and Line 232, Fig 5, Line 267 Fig 8 are a very low quality. Please provide better quality figures.

Line 278, the structure for micrococcin P1 should have a figure number and please improve the quality of the structure and the analogues.

Line 295, experimental section for the isolation of cyclic despeptide so I would expect to see 1H NMR and 13C spectra in the SI?

The authors mentioned many times that these metabolites possessed an antibiotic properties, however there is no mention in the text what type of biological assays have they done to assess the antibacterial properties of the extracts and/or pure compounds?

Line 277, The authors need to reisolate the micrococcin P1 and P2 and compare the NMR data to the reported one? To confirm the stereochemistry

Reviewer 2 Report

This manuscript describes the detection and isolation of 137 bacterial strains from one field collection of cyanobacteria Moorea producens.  The antimicrobial activity of organic extracts for 24 of them had been evaluated. Two known natural products, micrococcins P1 and P2 had been isolated from the most potent extract and identified by predicted high-resolution MS. Three additional known natural products had been identified by the high-resolution MS as well. Also, microscopic visualizations of multiple bacteria on the sheath of moorea allowed them to establish their positions beneath or above the sheath surface and discuss their ecological roles.

Across the manuscript, authors apply the name Moorea that had recently been renamed Moorena.

1. Tronholm, A. and Engene, N. (2019) Moorena gen. nov., a valid name for “Moorea Engene & al.” nom. inval. (Oscillatoriaceae, Cyanobacteria). Notulae Algarum 122, 1– 2 2. https://www.algaebase.org/search/genus/detail/?tc=accept&genus_id=52740&-session=abv4:AC1F06400314d2A9C2PnAC5F39F0

I did not find Apendix 1 the authors mention on line 284 but the Experimental part is missing how micrococcins had been isolated in this study. 

I was surprised that the identification of dolastatins is confirming Moorena identity in this study.

Reviewer 3 Report

The authors discussed of antibiotics produced by a Kenyan strain. They prove the production of micrococcin P1 by mass spectrometry and they studied all the population of bacteria linked to the cyanobacteria.

It could be interesting to have a figure that shows the minimal inhibitory concentration assays. These types of experiments are described in the methods but not illustrated in the result and discussion part. It could be good to add a part about this.